# Distributed Multi-Robot Information Gathering under Spatio-Temporal Inter-Robot Constraints[note 1]

**DOI:** 10.3390/s20020484

**Published:** 2020-01-15

**Authors:** Alberto Viseras, Zhe Xu, Luis Merino

**Affiliations:** 1German Aerospace Centre (DLR), 82234 Oberpfaffenhofen, Germany; 2Australian Centre for Field Robotics (ACFR), Sydney, NSW 2006, Australia; zhexu14@gmail.com; 3School of Engineering, Universidad Pablo de Olavide (UPO), 41013 Seville, Spain; lmercab@upo.es

**Keywords:** robotics, distributed multi-agent systems, information gathering, Gaussian processes

## Abstract

Information gathering (IG) algorithms aim to intelligently select the mobile robotic sensor actions required to efficiently obtain an accurate reconstruction of a physical process, such as an occupancy map, a wind field, or a magnetic field. Recently, multiple IG algorithms that benefit from multi-robot cooperation have been proposed in the literature. Most of these algorithms employ discretization of the state and action spaces, which makes them computationally intractable for robotic systems with complex dynamics. Moreover, they cannot deal with inter-robot restrictions such as collision avoidance or communication constraints. This paper presents a novel approach for multi-robot information gathering (MR-IG) that tackles the two aforementioned restrictions: (i) discretization of robot’s state space, and (ii) dealing with inter-robot constraints. Here we propose an algorithm that employs: (i) an underlying model of the physical process of interest, (ii) sampling-based planners to plan paths in a continuous domain, and (iii) a distributed decision-making algorithm to enable multi-robot coordination. In particular, we use the max-sum algorithm for distributed decision-making by defining an information-theoretic utility function. This function maximizes IG, while fulfilling inter-robot communication and collision avoidance constraints. We validate our proposed approach in simulations, and in a field experiment where three quadcopters explore a simulated wind field. Results demonstrate the effectiveness and scalability with respect to the number of robots of our approach.

## 1. Introduction

Information gathering (IG) is a key task in many robotic applications such as, e.g., magnetic field mapping [1], environmental monitoring [2], or wind field mapping [3]. The IG task can clearly benefit from distributed multi-robot coordination strategies: First, by means of parallelization, as tasks could be split between robots, and second, in terms of robustness, as tasks of a faulty robot could be overtaken by the rest of the team.

A common approach often used in the literature to solve multi-robot information gathering (MR-IG) tasks is to use an underlying model of the physical process under study that, together with an information-theoretic metric, is employed to predict the impact of certain robot actions and states (see, e.g., [1,4]). In particular, in this work we assume Gaussian processes (GPs) for regression [5] as underlying model, and mutual information (MI) [6] as information metric.

GPs are state-of-the-art models to represent spatio-temporal fields [1,2,3,5]. Furthermore, the use of GPs together with MI has been shown to significantly outperform former MR-IG algorithms [7,8]. However, most state-of-the-art algorithms employ discretization of the robots state and action spaces, and do not take into account robots’ dynamics. This makes such strategies computationally intractable for robotic systems with complex dynamics, like, e.g., an aircraft. This limits the applicability of state-of-the-art algorithms to a reduced class of "simple" robots. In addition, most MR-IG algorithms do not take into account inter-robot constraints like, e.g., inter-robot collision avoidance or communication constraints. Such limitations preclude state-of-the-art algorithms to be transferred from simulations to real world experiments.

In this paper we propose an algorithm that tackles the two aforementioned issues: generality of the strategies, and handling of inter-robot constraints. Our algorithm consists of two major blocks. First, we use rapidly-exploring random trees (RRT) to design an algorithm that can be generalized to multiple classes of robots, as RRT is a state-of-the-art strategy to plan paths in continuous high dimensional spaces. Second, we employ distributed constraint optimization (DCOP) techniques to handle inter-robot constraints in a distributed fashion. In particular, we employ Max-sum for multi-robot coordination [9]. Max-sum, in contrast to other DCOP techniques (see [9] for an overview), makes efficient use of the computational and communication resources. In addition, it offers approximate solutions that are close to optimal for many applications of interest like, e.g., exploration, or tracking [4], as well as a solution for cooperative games [10] that can be also used to model multi-robot cooperation for IG tasks [11].

In this paper, we build on RRT and max-sum, together with GPs and MI, to derive a novel active perception strategy that allows multiple robots to autonomously gather information of a physical process of interest. Specifically, in opposition to state-of-the-art strategies (see Section 2), our proposed strategy is able (i) to account for robots with complex dynamics that operate in a continuous environment; (ii) to solve an IG task in a distributed fashion with local inter-robot communication; and (iii) to incorporate spatial and temporal inter-robot constraints.

We evaluate our approach in simulations where multiple aerial vehicles, which are subject to collision avoidance and communication constraints, cooperate to map a wind field. In addition, we carry out an outdoor field experiment where a fleet of three aerial robots explore a simulated wind field. Results of the experiment demonstrate the effectiveness, as well as the online realization of the algorithm.

Let us also point out that the work presented here is largely based on our previous publication [12], yet it extends the latter in several important respects. In particular,

We introduce a discussion on state-of-the-art information metrics for IG. This discussion allows us to motivate and proof, empirically through simulations, our specific choice.We introduce algorithmic approximations that permit an online realization of our algorithm. In addition, we present a detailed analysis of our algorithm’s computational complexity.We include an inter-robot collision avoidance constraint that permits collision-free IG.We extend the simulations setup from four to up to eight robots. This allows us to show the algorithm’s scalability as the number of robots increases.We analyze in detail the effect of algorithm parameters in the algorithm’s performance.We provide a full description of the experimental setup, and additional experimental results.

The remainder of the paper is organized as follows. First, we review the related work in Section 2. Then we state the problem formally in Section 3. Next we introduce in Section 4 the methods in which our algorithm builds. Our algorithm relies on an information metric to decide robots’ action. Therefore, we include in Section 5 a discussion about the suitability of several information metrics for our specific problem. This is followed in Section 6 and Section 7 by a detailed explanation of the different subsystems that compose our system, and an assessment of the algorithm’s computational complexity, respectively. Then we test the algorithm in simulations in Section 8, and verify in Section 9 its performance with a field experiment in which three quadcopters explore an unknown simulated wind field. We finalize with a summary and outlook of the paper in Section 10.

## 2. Related Work

Multi-robot IG is a topic that has attracted lots of interest in the last years, and many strategies have been proposed recently. Here we focus on model-based strategies that use GPs as underlying model of a process of interest. In [6] the authors present a multi-robot exploration algorithm that exploits the properties of GPs as underlying model and of MI as information metric. By leveraging the submodularity property of MI, the authors derive a sequential greedy algorithm that offers worst case guaranties. However, the algorithm proposed in [6] is centralized, as it requires robots to have access to information from the complete team.

A solution that goes beyond a centralized architecture was proposed in [13], where the authors propose a strategy that is semi-decentralized. That is, the algorithm combines a centralized and a decentralized processing: On the one hand, a master robot selects a set of observation points that maximize a joint entropy. On the other hand, robots perform data fusion in a decentralized fashion.

In contrast to [6,13], here we argue for the use of a decentralized architecture to decide robots movement. Multiple works proposed decentralized IG methods with GPs; see, e.g., [1,4,13,14]. The previous techniques [1,4,13,14] employ a discretization of the robot state and action spaces and search-based planning algorithms for exploration. Thus, they cannot consider kinematic constraints associated to the robot motion for non-holonomic vehicles, like fixed-wing aircrafts.

Informative path planning techniques typically encompass algorithms that aim to plan a path which is both feasible, given a robot’s dynamical constraints, and optimal with respect to some information quality metric. Single-robot approaches for informative path planning in continuous spaces have been proposed in [15,16,17,18,19,20]. However there is little work in the literature that propose multi-robot informative path planning algorithms. In [17] the authors present a multi-robot sampling-based informative path planner. However, [17] is limited to the particular case of tracking applications. In [21] a multi-robot information gathering method is proposed, which is also designed for tracking applications.

Extending the informative path planning problem to multi-robot settings involves two key tasks. First, robots must be able to cooperate to maximize the fleet’s information gain. Furthermore, robots must be able to handle mission-specific spatio-temporal inter-robot constraints. To tackle the two aforementioned tasks, [8] and [22] divide the problem in task and path planning. In particular, [8] and [22] consider at task level GPs and a MI utility function to determine informative points to be visited in a multi-robot exploration scenario. Then, paths are planned towards those points in a centralized way. In contrast, we propose in this paper an algorithm that allows robots to plan paths in a decentralized fashion using inter-robot local communication.

One fundamental aspect in multi-robot IG is how to handle inter-robot spatio-temporal constraints. In [23] the authors propose a method to handle such constraints for a target search problem. Specifically, they employ the augmented Lagrangian method for the multi-robot cooperation. The use of the Lagrangian methods typically requires constraints and the objective function, which guides robots movement, to be differentiable. This limits the applicability of the algorithm to a selected class of objective functions and constraints, and limits the generality of the algorithm. On the contrary, this paper presents a method that is able to incorporate a large class of objective functions; the only requirement is that the global objective function can be expressed as the sum of the individual objective functions of each of the agents. To this end, we employ the max-sum algorithm from [24], and extend it to allow exploration in a continuous space for robots with kinematic, kinodynamic and spatial-temporal constraints.

## 3. Problem Statement

We consider here a problem of exploring an *a priori* unknown physical process with *N* cooperative robots autonomously and as accurately as possible, in the sense of minimizing the Root Mean Squared Error (RMSE) between a process estimate (given by a GP model) and the (unknown) ground truth. Additionally, exploration should be efficient, in the sense of minimizing the RMSE as fast as possible, provided the (i) available resources, and (ii) complex constraints such as inter-robot collision avoidance and communication constraints.

To achieve this, we make a few simplifying assumptions. Specifically, we assume the following:The physical process of interest can be modeled with a GP sufficiently well.This process is time-invariant during the IG task.The robot positions are known exactly and are noise-free. That is, we assume that there exists an external positioning system that provides us with a highly accurate localization, e.g., a Real-Time Kinematic navigation for global positioning systems (GPS-RTK) for outdoor scenarios, or a motion tracking system for indoor environments. Uncertainty in positioning could also be accounted for using GPs [25], but it is out of the scope of this work.

Additionally, our problem is subject to the following physical constraints:4.The robot *i*, with i=1,…,N, motion model is given by a known function $x_{i}\left(t+\Delta t\right)=f\left(x_{i}\left(t\right),u_{i}\right)$ that relates the robot’s current position $x_{i}\left(t\right)$ and future position $x_{i}\left(t+\Delta t\right)$ given a control input $u_{i}$, where Δt is the duration of a single time step.5.Robots can only directly communicate if they are neighbors, i.e., if they are separated by less than a distance $r_{c}$. This defines a robots communication graph $G_{c}\left(V_{t},E_{t}\right)$ at time *t*, with $V_{t}=\left\{x_{1}\left(t\right),\ldots ,x_{N}\left(t\right)\right\}$, and $E_{t}=\left\{\left(x_{i}\left(t\right),x_{j}\left(t\right)\right):i,j\in \left\{1,\ldots ,N\right\},i\neq j,||x_{i}\left(t\right)-x_{j}\left(t\right)\left|\right|\leq r_{c}\right\}$. We also assume that there exists an underlying communication protocol, like, e.g., TCP/IP, which ensures an error-free data transmission.

Furthermore, robots must fulfill the following mission-related constraints:6.*Inter-robot collision avoidance*: two robots collide if they are separated less than a distance $r_{s}$. Distance $r_{s}$ shall take into account robots’ shape, as well as the potential uncertainty in robot positions.7.*Network connectivity*: the network of robots requires a periodic connectivity, with a maximum disconnection time of $k_{c}\Delta t$ seconds, where $k_{c}$ is a constant that denotes a number of time steps.

Let us now introduce some notation that we will use in the remainder of the paper. The position of robot *i* will be denoted by $x_{i}\left(t\right)\in X_{free}$, where $X_{free}\in R^{d_{s}}$ corresponds to the free space in the robot’s configuration space, with $d_{s}$ being the dimensionality of the environment in which a process of interest takes place. The physical process at position $x\in X_{free}$ is denoted as y(x)∈R. Typically, however, a process is not observed directly, but is measured using some sensors. Here we assume a simple sensor model that represents a measured process as z(x)=y(x)+ϵ(x), where z(x) is a process sample, y(x) is the unobserved true process value, and ϵ(x) is a random noise. In the following we will assume that, for different measurements, noise samples ϵ(x) are independent and identically distributed according to $N(0,\sigma_{n}^{2})$; i.e., they follow a Gaussian distribution with zero mean and variance $\sigma_{n}^{2}$.

The afore-described problem requires an infinite number of steps to be completed, which corresponds to an infinite horizon IG task. However, for the sake of computational feasibility, a common approach in IG is to divide an infinite horizon problem into multiple finite horizon problems that can be solved individually and sequentially. In particular, here we assume an horizon of $k_{c}$ time steps. That is, every $k_{c}$ time steps robots solve an IG problem to find a set of paths $P=\left\{P_{1},\ldots ,P_{N}\right\}$, with $P_{i}\subset X_{free}$, i=1,…,N, which maximizes a global utility function $U_{I}\left(\cdot ,\cdot \right)$. The utility function $U_{I}\left(P,X\right)$ depends on the paths P, and on measurements already gathered by robots at positions contained in matrix $X\subset X_{free}$. Additionally, robots must fulfill physical and mission related-constraints. Once robots find a solution to the finite horizon IG problem, robots follow $P_{i}$ while taking measurements along it, and repeat the procedure again.

More formally, in this paper we propose an approximate solution to the following finite horizon problem:(1)$$\begin{array}{cc} \underset{P}{\mathrm{maximize}}\; & U_{I}\left(P,X\right)\\ \mathrm{subject}\;\mathrm{to}\; & P_{i}=\left\{x_{i}\left(t+dt\right),\ldots ,x_{i}\left(t+k_{c}dt\right)\right\}\\ \; & x_{i}\left(t+k_{t}dt\right)=f\left(x_{i}\left(t+\left(k_{t}-1\right)dt\right),u_{i}\right),\\ \; & \begin{array}{c} \left|\right|x_{i}\left(t+k_{t}dt\right)-x_{j}\left(t+k_{t}dt\right){\left|\right|}_{2}^{2}\geq r_{s}, \end{array}\\ \; & G_{c}\left(V_{t+k_{c}dt},E_{t+k_{c}dt}\right)\;\mathrm{is}\;\mathrm{connected}. \end{array}$$
with $k_{t}=1,\ldots ,k_{c}$, i,j=1,…,N and i≠j, and $P_{i}\subset P$.

To solve problem (Equation 1) we build on three main methods: GPs, RRT, and max-sum algorithm. Next we give an overview of the three methods.

## 4. Background

### 4.1. Gaussian Processes for Modelling Spatial Data

A GP is a collection of random variables, any finite number of which have a joint multivariate Gaussian distribution [5]. A GP is defined by m(x), the mean function, and by $k(x,x^{\prime },\theta )$, the covariance function, over positions x, $x^{\prime }$. Here we assume a zero mean prior function m(x), which implies an absence of a priori known values of the process. The covariance function depends on hyperparameters θ. We use the squared exponential (SE) [5] covariance function due to its capacity to model smooth processes. This function is determined by hyperparameters $\theta ={\left[\sigma_{f}^{2},l,\sigma_{n}^{2}\right]}^{T}$, being *l* the characteristic length-scale (informally, "how close" two positions x and $x^{\prime }$ have to be to influence each other significantly); $\sigma_{f}^{2}$ represents the maximum allowable covariance; and $\sigma_{n}^{2}$ is the variance of the noise fluctuations [5].

We use the following definitions: $X={\left[x^{\left[1\right]},x^{\left[2\right]},\cdots ,x^{\left[n\right]}\right]}^{T}$ is a matrix where each row corresponds to a spatial location where a robot has gathered a measurement. Vector $z={\left[z^{\left[1\right]},z^{\left[2\right]},\cdots ,z^{\left[n\right]}\right]}^{T}$ stores the corresponding measurements. And in matrix $X_{*}={\left[x_{*}^{\left[1\right]},x_{*}^{\left[2\right]},\cdots ,x_{*}^{\left[p\right]}\right]}^{T}$ each row is a “probe” location – points in space where we predict the process value using the GP model. In this paper, “probe” locations correspond to points where robots aim to potentially take a measurement. Moreover, we define matrices $K,K_{*},K_{**}$ from the covariance function $k(x,x^{\prime },\theta )$ as follows: (2)$$K=\left[\begin{array}{ccc} k(x^{\left[1\right]},x^{\left[1\right]}) & \cdots & k(x^{\left[1\right]},x^{\left[n\right]})\\ \vdots & \ddots & \vdots \\ k(x^{\left[n\right]},x^{\left[1\right]}) & \cdots & k(x^{\left[n\right]},x^{\left[n\right]}) \end{array}\right],\;K_{*}=\left[\begin{array}{ccc} k(x^{\left[1\right]},x_{*}^{\left[1\right]}) & \cdots & k(x^{\left[1\right]},x_{*}^{\left[p\right]})\\ \vdots & \ddots & \vdots \\ k(x^{\left[n\right]},x_{*}^{\left[1\right]}) & \cdots & k(x^{\left[n\right]},x_{*}^{\left[p\right]}) \end{array}\right],\;K_{**}=\left[\begin{array}{ccc} k(x_{*}^{\left[1\right]},x_{*}^{\left[1\right]}) & \cdots & k(x_{*}^{\left[1\right]},x_{*}^{\left[p\right]})\\ \vdots & \ddots & \vdots \\ k(x_{*}^{\left[p\right]},x_{*}^{\left[1\right]}) & \cdots & k(x_{*}^{\left[p\right]},x_{*}^{\left[p\right]}) \end{array}\right].$$

Let us emphasize that K, $K_{*}$, and $K_{**}$ are all functions of θ through k(·). We do not include this dependency to simplify notation.

From measurements z at positions X, we can predict the process values $y_{*}$ at locations $X_{*}$ and the associated uncertainties. Vector $y_{*}$ is a random vector with the following conditional distribution: $p(y_{*}|X_{*},X,z)=N$$\left(\mu_{*},\Sigma_{*}\right)$, where $\mu_{*}$ and $\Sigma_{*}$ are computed as (see [5] for more details):(3)$$\begin{array}{cc} \mu_{*} & =m\left(X_{*}\right)+K_{*}^{T}K^{-1}\left(z-m\left(X\right)\right),\\ \Sigma_{*} & =K_{**}-K_{*}^{T}K^{-1}K_{*}. \end{array}$$

Learning a GP model implies estimating the value of the hyperparameters $\theta_{*}$ that best fit the measurements z at locations X. This estimation is generally formulated as a maximum-likelihood problem, where the log-marginal likelihood (LML) with respect to θ is maximized:(4)$$\theta_{*}=\underset{\theta }{\mathrm{argmax}}\left\{-\frac{1}{2}z^{T}K^{-1}z-\frac{1}{2}\log\left|K\right|\right\}.$$

This is a nonlinear optimization problem that requires application of numerical optimization techniques [5].

### 4.2. Rapidly Exploring Random Trees

The RRT algorithm allows robots to plan paths in complex high dimensional spaces [26]. The RRT algorithm iteratively constructs a graph G(V,E) (tree) with a set of vertices V and edges E with the goal of finding possible trajectories starting from a state $x_{A}$.

The algorithm is realized as follows: It draws a sample $x_{rand}$ randomly from a uniform distribution defined over $X_{free}$. Then it finds the nearest neighbor $x_{nearest}$ (in terms of the cost-to-reach) of $x_{rand}$ in the set of vertices V. Next it simulates driving the robot from $x_{nearest}$ to $x_{rand}$ according to the robot’s controller. In particular, it drives the robot a maximum distance η, which is a user-selected parameter that sets the maximum branch size. This results in a new state $x_{new}$. If trajectory $E(x_{nearest},x_{new})$ does not collide with any obstacles, it adds vertex $x_{new}$ and edge $E(x_{nearest},x_{new})$ to tree G. This process is repeated during $N_{p}$ iterations.

### 4.3. Max-Sum Algorithm

Let us consider a team of *N* robots, where each robot *i* can control a decision variable $D_{i}$ that can take values from domain $C_{i}=\left\{C_{i}^{\left[1\right]},C_{i}^{\left[2\right]},\ldots ,C_{i}^{\left[k_{i}\right]}\right\}$. In this paper, $C_{i}^{\left[i^{\prime }\right]}\subset C_{i}$ with $i^{\prime }=1,\ldots ,k_{i}$ consists of a set of potential measurement locations that robot *i* could visit. We denote the set of variables for which we aim to solve the assignment problem as $D=\{D_{1},D_{2},\ldots ,D_{N}\}$. For example, for a team of three robots with identical domain size $k_{i}=4$ with i=1,2,3, a possible assignment for the variables could be $D=\{D_{1}:C_{1}^{\left[1\right]};\;D_{2}:C_{2}^{\left[4\right]};\;D_{3}:C_{3}^{\left[1\right]}\}$.

In max-sum [9], the goal of the robots is to maximize a global utility function $U\left(D\right)=\sum_{i=1}^{N}U_{i}\left(\bar{D_{i}}\right)$, where $U_{i}\left(\bar{D_{i}}\right)$ denotes utility function of robot *i*, and $\bar{D_{i}}\subset D$. For instance: coming back to the previous three robots example, $\bar{D_{1}}=\left\{D_{1},D_{2}\right\}$ implies that the utility of robot 1 depends on its own decision, and on the decision of robot 2, but not on the one from robot 3. Within this setting, we wish to find the optimal assignment $D^{*}$ such that U(D) is maximised: $D^{*}=\underset{D}{\mathrm{argmax}}\sum_{i=1}^{N}U_{i}\left(\bar{D_{i}}\right)$.

Max-sum formulates this assignment problem as a *factor graph* [27]. A factor graph is a bi-partite graph with two types of nodes: variables and factors. Edges in this graph represent the dependencies of factors on variables. For instance, the factor graph in Figure 1 represents $U\left(D\right)=U_{1}\left(\bar{D_{1}}\right)+U_{2}\left(\bar{D_{2}}\right)+U_{3}\left(\bar{D_{3}}\right)$, where $\bar{D_{1}}=\left\{D_{1},D_{2}\right\}$, $\bar{D_{2}}=\left\{D_{1},D_{2},D_{3}\right\}$ and $\bar{D_{3}}=\left\{D_{2},D_{3}\right\}$.

Max-sum is a message passing algorithm on factor graphs. Messages are passed along the edges of the factor graph in order to determine the variable values that maximise U(·). We distinguish between two types of messages:Factor to variable message; denoted $s_{i\to j}$. It is the maximum value of factor $U_{i}$ for each possible value of $D_{j}$.Variable to factor message; denoted $q_{j\to i}$. It is the maximum value of $U_{i}$ neighboring factors for each possible value of $D_{j}$.

For more details on the definition of messages we refer the reader to the original paper [9]. Provided the definitions of messages we can now summarize the execution of max-sum algorithm. First, each of the robots arbitrarily initializes $q_{j\to i}$ and sends it to its adjacent function nodes. This triggers an exchange of messages between variable and utility function nodes. The messages exchange will continue until message values converge, or after an user-defined number of iterations. Next, each of the robots evaluates the marginal function of variable $D_{i}$: $z_{i}\left(D_{i}\right)$ [9]. Then, by simply finding ${\mathrm{argmax}}_{D_{i}}z_{i}\left(D_{i}\right)$, each individual robot *i* is able to determine which $C_{i}^{\left[i^{\prime }\right]}$, for $i^{\prime }=1,2,\ldots ,k_{i}$, it should visit such that U(·) is maximized.

Max-sum algorithm delivers an exact solution in cases where the factor graph is acyclic, i.e., it has no loops. Otherwise, if the factor graph is cyclic, i.e., it has loops (as in, e.g., Figure 1), max-sum has been empirically shown to converge to an approximation of the exact solution [9]. Moreover max-sum is robust against communication delays. Since max-sum messages are transmitted asysnchronously and do not follow a pre-defined order, max-sum is resilient to delays in the data transfer.

## 5. Information Metric

Information metrics are used by robots to guide their movement by selecting positions $X_{*}$ that maximize a particular information metric. Here we compare information metrics in the context of GPs based on two properties that are useful for IG. These two properties are:*monotonicity* as we increase the number *p* of potential measurement locations $X_{*}$. That is, we are interested in information metrics that yield a higher value as we consider longer paths, i.e., paths with a higher *p*; and*submodularity* respect to *p*. In short, a submodular information metric offers diminishing returns as we increase *p*. This justifies the use of finite horizon approaches (as we do in this paper), as the amount of information obtained by increasing *p* becomes irrelevant from a certain value of *p*. For a detailed overview of submodularity applications in the context of GPs, we refer the reader to [28].

We analyze three information metrics: (i) Differential Entropy, (ii) Mutual Information Non-Measured, and (iii) Mutual Information All. To better support this analysis, we include in Figure 2 a graphical representation of some basic notation employed in this paper.

**Differential Entropy.** We denote the differential entropy of a process given by random variable $Y_{X_{*}}$, defined at potential measurement locations $X_{*}$, as $H(Y_{X_{*}}|X)$, with X the location of measurements gathered by the robot up to now. $H(Y_{X_{*}}|X)$ can be calculated with $H\left(Y_{X_{*}}|X\right)=\frac{1}{2}\log({\left(2\pi e\right)}^{p}|\Sigma_{*}\left|\right)$, with $\Sigma_{*}$ calculated with (Equation 3).

**Mutual Information Non-Measured.** We define Mutual
Information
Non-Measured as the MI between: a random variable $Y_{X_{*}}$; and a random variable $Y_{V_{X_{free}}\backslash \left\{X\cup X_{*}\right\}}$ that represents the physical process at $V_{X_{free}}$ that would remain unmeasured after visiting $X_{*}$. Note that we employ here $V_{X_{free}}$ instead of $X_{free}$ because MI for GPs is evaluated at a set of discrete locations [28]. Therefore, to calculate MI we discretize $X_{free}$ by overlaying a lattice graph with vertices $V_{X_{free}}$. Also note that, for $x\in X,X_{*}$ and $x^{\prime }\in V_{X_{free}}$, we assume that $x=x^{\prime }$ if x lies within the cell associated to $x^{\prime }$ (see Figure 2).

Mutual Information Non-Measured is given by: $I\left(Y_{V_{X_{free}}\backslash \left\{X\cup X_{*}\right\}};Y_{X_{*}}|X\right)=H\left(Y_{V_{X_{free}}\backslash \left\{X\cup X_{*}\right\}}|X\right)$$-H(Y_{V_{X_{free}}\backslash \left\{X\cup X_{*}\right\}}|X,Y_{X_{*}})$. This expression has a clear interpretation for IG: we aim to sample at locations $X_{*}$ that yield a maximum inter-dependence with process $Y_{V_{X_{free}}\backslash \left\{X\cup X_{*}\right\}}$, defined at all positions in the environment that will remain unmeasured.

**Mutual Information All.** In this chapter we propose the use of Mutual
Information
All. This calculates the MI between a random variable $Y_{X_{*}}$; and a random variable $Y_{V_{X_{free}}}$. Mutual Information All is given by the following expression: $I\left(Y_{V_{X_{free}}};Y_{X_{*}}|X\right)=H\left(Y_{V_{X_{free}}}|X\right)-H\left(Y_{V_{X_{free}}}|X,Y_{X_{*}}\right)$. This metric is similar to Mutual
Information
Non-Measured, but it has an interesting property that we discuss next.

The three afore-described information metrics are submodular [29]. To study the metrics’ monotinicity, we carried out a simple simulation. Specifically, we considered a one-dimensional space $V_{X_{free}}$ that consists of 90 equally separated positions. Then we assumed that a robot already took ten measurements, drawn from a GP at positions X randomly selected from $V_{X_{free}}$. For this setup, we evaluated the afore-described information metrics as we increase *p* (illustrating longer planing horizons). That is, we randomly selected from $V_{X_{free}}$ a number of potential measurements positions, which is given by $X_{*}$. Results from this experiment are depicted in Figure 3, where each dot corresponds to a realization of the experiment.

From Figure 3 we can draw the following conclusion: Differential Entropy is non-monotonic, which goes against the principle of “information never hurts”. Non-monotonocity is a property that is particularly undesirable for algorithms that aim to plan over an horizon longer than one step, as it is the case in Equation (Equation 1).

In addition to entropy, we analyzed two uses of MI: Mutual
Information
Non-Measured and Mutual
Information
All. According to Figure 3, Mutual
Information
Non-Measured is only monotonic in the first part of the curve. This implies that Mutual
Information
Non-Measured does not allow us to plan over arbitrarily long horizons, as longer paths may result in a lower value of the information metric. Note that this property goes again against the principle of “information never hurts”.

In contrast, here we propose the use of Mutual
Information
All as information metric to tackle this problem. Mutual
Information
All is monotonic (see Figure 3c), which is an ideal choice for IG tasks.

## 6. Distributed Multi-Robot Information Gathering Algorithm

We present in this section the algorithm that we propose to obtain an approximate solution of Equation (Equation 1). Our proposed algorithm works as follows: first, each robot plans a set of potential paths $P_{i}$ that it could follow (Section 6.1). Specifically, each robot generates a RRT, whose root is the robot’s current position. Next, robots cooperate in order to select a path that maximizes $U_{I}\left(\cdot ,\cdot \right)$ subject to physical and mission-related constraints from Equation (Equation 1). Here $U_{I}\left(\cdot ,\cdot \right)$ corresponds to Mutual
Information
All, as indicated in Section 5. To solve this multi-robot cooperation problem we propose the use of a DCOP algorithm: max-sum [9].

Max-sum requires that each robot knows its own set of potential paths, as well as its neighbors’ set of potential paths. This set of paths we term robot’s domain. To this end, we include a module that allows robots to find its neighbours, and to send its domain (Section 6.2).

Once a robot receives its neighbors’ domain, it executes Max-sum (Section 6.3). Max-sum allows us to solve a combinatorial optimization problem [9]. To solve the optimization problem, each robot must evaluate all combinations of potential paths from the received domains (including its own domain). As we previously mentioned, here we consider a robot’s domain as the set of all paths that are contained in the generated RRT. Since RRTs could grow large, the number of total paths could increase as well. This would result in an increase of the complexity of the combinatorial optimization, which could make the optimization computationally intractable. To solve this issue, we propose a procedure in which each robot groups the RRT paths into clusters, which reduces the robots’ domain size (Section 6.2).

Max-sum outputs a cluster for each individual robot that solves (Equation 1). Then, each robot selects a path within its cluster. This is realized by evaluating Mutual
Information
All (Section 6.4).

Next, robots follow the selected paths while taking measurements along them. Measurements encode the knowledge robots have about the process of interest. Therefore, they exchange the gathered measurements through the network; i.e., they perform data fusion (Section 6.5). Finally, robots update their GPs model with the new measurements in order to improve the process model (Section 6.6).

In Figure 4 we depict a block diagram of the proposed algorithm. In particular, the diagram corresponds to the modules that each single robot executes. Modules are executed in a loop, where each loop iteration solves Equation (Equation 1). Next we explain the algorithm’s modules in detail.

### 6.1. Calculate Candidate Paths and Generate Clusters

The first step of the algorithm is the computation of a set of feasible paths given $x_{i}$ and f(·). This is realized with the kinodynamic RRT algorithm. We introduce a constraint in RRT that guarantees collision-free paths between robots that cannot directly communicate with each other. We realize this by limiting the RRT planning horizon to a maximum distance of $(r_{c}-r_{s})/2$, with $r_{c}$ and $r_{s}$ the communication and safety radius, respectively (see constraints 5 and 6 in Section 3).

Let us denote the set of paths generated by robot *i* with RRT as $P_{i,rrt}$ (see Figure 5a). Ideally, we would like robots to exchange $P_{i,rrt}$, and calculate $P_{i}\in P_{i,rrt}$ that solves (Equation 1). However, as we pointed out in Section 6 introduction, this would translate in evaluating multiple combinations of paths, which is computationally intractable. Therefore, we introduce the concept of spatio-temporal clusters. In Figure 5 we illustrate the clustering procedure with an example.

Spatio-temporal clusters give us flexibility to adapt our algorithm to the robot’s computational capabilities: as we increase the number of spatial and temporal divisions, we get closer to the actual RRT. However, clusters may lead to a loss of performance when optimizing U(·), since robots combine several paths into a cluster during the cooperation procedure. Nevertheless, we demonstrate in Section 8.5 that performance decrease is negligible for a sufficiently large number of clusters (approx. 18 clusters for our setup).

Next we explain the clustering procedure in detail. First, we define $k_{t}$ temporal horizons. Temporal horizons represent time spans of a path, which are given by the robot’s motion model. For each of the temporal horizons, we extract the corresponding paths from the RRT. Then, we group paths of equal temporal horizon into $k_{s}$ spatial clusters. This last step is realized running the k-means technique [30] over the complete paths.

The clustering procedure is executed by each of the robots individually, yielding $k_{t}\times k_{s}$ clusters for each robot. We denote the clusters of robot *i* by $C_{i}=\left\{C_{i}^{\left[1\right]},C_{i}^{\left[2\right]},\ldots ,C_{i}^{\left[k_{t}k_{s}\right]}\right\}$. Note that $C_{i}^{\left[j\right]}\subset P_{i,rrt}$ for all $j=1,\ldots ,k_{t}k_{s}$. Let us recall that, according to notation introduced in Section 4.3, $C_{i}$ corresponds to robot *i* domain.

### 6.2. Search Neighbors and Exchange Domains

Robots move as they explore the process of interest, which results in a variation of the network topology. Therefore, we introduce a neighbors search mechanism. Robots realize this by sending an identification message with its ID. Robots that receive the identification message (see robots communication model defined in constraint 5, Section 3) add the corresponding robot’s ID to its set of neighbors. Then each robot i=1,2,…,N sends $C_{i}$ to their neighbors. The sharing of domains is the input needed to initiate the multi-robot cooperation procedure, which we explain next in detail.

### 6.3. Calculate Robot Utilities and Execute Max-Sum

Once robots exchange domains, they execute a max-sum algorithm to perform an assignment of clusters that approximately solves (Equation 1). Specifically, robots cooperate to find the individual cluster $D_{i}:C_{i}^{\left[i^{\prime }\right]}\in C_{i}$, with $i^{\prime }=1,2,\ldots ,k_{i}$, that each robot should select in order to maximize a global utility function U(·). Here we define U(·) so that it consists of two terms:an *information gathering* term, denoted as ${\tilde{U}}_{I}\left(D,X\right)$, which measures the informativeness of a particular assignment of clusters D given previously collected measurements X. Note that here we refer to ${\tilde{U}}_{I}\left(\cdot \right)\approx U_{I}\left(\cdot \right)$ as robots aim to find a joint cluster assignment, instead of a joint assignment of paths as in (Equation 1); anda *constraint satisfaction* term, denoted as $U_{C}\left(D\right)$, which enforces problem (Equation 1) constraints.

The combination of these terms yields our proposed utility function:(5)$$U\left(D,X\right)={\tilde{U}}_{I}\left(D,X\right)-U_{C}\left(D\right).$$

Let us next describe ${\tilde{U}}_{I}\left(\cdot \right)$, $U_{C}\left(\cdot \right)$ in more detail.

#### 6.3.1. Information Gathering Utility—${\tilde{U}}_{I}\left(D,X\right)$

We define ${\tilde{U}}_{I}\left(\cdot \right)$ as the MI between $Y_{V_{X_{free}}}$, and a joint assignment of clusters D, conditioned on X. This corresponds to Mutual Information All, described in Section 5, and is given by the following expression: ${\tilde{U}}_{I}\left(D,X\right)=I\left(Y_{V_{X_{free}}},Y_{D_{1}},Y_{D_{2}},\ldots ,Y_{D_{N}}|X\right)$, with $Y_{D_{i}}$ a GP that represents y(x) for all $x\in D_{i}$, i=1,2,…,N.

Our goal is to maximize ${\tilde{U}}_{I}\left(\cdot \right)$ in a decentralized fashion. To this end, we employ max-sum algorithm, and we express ${\tilde{U}}_{I}\left(\cdot \right)$ as a sum of functions that are associated to each individual robot (see Section 4.3). By applying the chain rule for MI, and decomposing $I(Y_{V_{X_{free}}},Y_{D_{1}},Y_{D_{2}},\ldots ,Y_{D_{N}}|X)$ as a difference of conditional entropies, we can express ${\tilde{U}}_{I}\left(\cdot \right)$ as:(6)$$\begin{array}{cc} {\tilde{U}}_{I}\left(D,X\right) & =I\left(Y_{V_{X_{free}}},Y_{D_{1}},Y_{D_{2}},\ldots ,Y_{D_{N}}|X\right)=\sum_{i=1}^{N}I\left(Y_{V_{X_{free}}},Y_{D_{i}}|Y_{D_{i+1}},\ldots ,Y_{D_{N}},X\right)\\ \; & =\sum_{i=1}^{N}H\left(Y_{V_{X_{free}}}|Y_{D_{i+1}},\ldots ,Y_{D_{N}},X\right)-H\left(Y_{V_{X_{free}}}|Y_{D_{i}},\ldots ,Y_{D_{N}},X\right). \end{array}$$

Equation (Equation 6) cannot be directly applied for a decentralized system that relies on local communication between robots, as robot *i* domain only contains information about robot *i* neighbors; not about all robots with a higher ID (as required in Equation (Equation 6)). We solve this issue by applying the principle of *locality* [13,24]. This allows us to assume that two random variables $Y_{D_{j}},Y_{D_{k}}$ are statistically independent if spatial locations contained in $D_{j},D_{k}$ are sufficiently distant. For our applications of interest, the principle of locality is a reasonable assumption as a process spatial correlation is typically much smaller than robots communication range. For example, in this paper’s motivating problem of mapping a wind field, the structures (thermals) are only a few hundred meters in size. In contrast, the robots communication range tend to be in the order of kilometers.

By considering the *locality* assumption we can now formulate Equation (Equation 5) as:(7)$$\begin{array}{c} U\left(D,X\right)=\sum_{i=1}^{N}H\left(Y_{V_{X_{free}}}|Y_{N(D_{i+1:N})},X\right)-H\left(Y_{V_{X_{free}}}|Y_{D_{i}},Y_{N(D_{i+1:N})},X\right)-U_{C}\left(D_{i},N\left(D_{i}\right)\right), \end{array}$$
where $N(D_{i+1:N})$ denotes assignment variables associated just to the neighbors of robot *i* with a higher ID, and $N(D_{i})$ denotes assignment variables that are associated to neighbors of the i-th robot.

#### 6.3.2. Constraint Satisfaction Utility—$U_{C}\left(D\right)$

The role of $U_{C}\left(\cdot \right)$ is to satisfy that mission-related constraints (constraints 6, 7 in Section 3) are not violated. To this end, we set $U_{C}\left(\cdot \right)=0$ if robots are in a configuration that is far from violating the constraints. Otherwise, we set $U_{C}\left(\cdot \right)$ to a value that increases within a “escape” distance $r_{e}$ as robots get closer to a configuration where constraints could be violated. In case a constraint is violated we set $U_{C}\left(\cdot \right)=\infty$ (see Figure 6).

Next we describe how we account for problem specific constraints:*Inter-robot collision avoidance*: we penalize robots that are separated a distance smaller than $r_{s}+r_{e}$.*Periodic network connectivity*: we penalize robots configurations that could lead to a disconnected network at $t+k_{c}\Delta t$; i.e., at the end of robots’ paths.

As robots rely on local communication, it is not trivial to guarantee a periodic network connectivity [31]. Therefore, as in [23], we guarantee connectivity by forcing robots to form a minimal topology – chain topology. That is, we encourage robots to be at least connected to their peers that have an immediate lower and higher ID. This way, we can solve the communication constraint only with local communication. Note that in our proposed approach more complex mechanisms like e.g., [31,32] could be introduced in the cooperation procedure to guarantee network connectivity.

#### 6.3.3. Path Selection

Equation (Equation 7) can be optimized in a decentralized fashion using max-sum algorithm (see Section 4.3). In our case, max-sum outputs for each robot *i*, i=1,2,…,N, an optimal cluster $D_{i}^{*}\in C_{i}$. $D_{i}^{*}$ is a cluster that contains multiple paths. Therefore, robots must select a path $P_{i}^{*}$ to follow from $D_{i}^{*}$. This is done by calculating the MI between $Y_{V_{X_{free}}}$, and a random variable $Y_{P_{i}}$ that represents all possible path assignments within the selected cluster, with $P_{i}\in D_{i}^{*}$. We condition MI on the knowledge about the selection of clusters ${\left\{D_{j}^{*}\right\}}_{j\in N_{i}}$ of neighboring robots $N_{i}$, and previously gathered measurements X. More formally, each robot *i* aims to find $P_{i}^{*}$ such that $P_{i}^{*}=\underset{P_{i}}{\mathrm{argmax}}I\left(Y_{V_{X_{free}}},Y_{P_{i}}|{\left\{D_{j}^{*}\right\}}_{j\in N_{i}},X\right)$. Let us remark that this procedure is done by each of the robots independently.

### 6.4. Follow Path and Collect Measurements

The output of the cooperation stage is a path $P_{i}^{*}$. Then robots follow $P_{i}^{*}$, and collect measurements along it. Robots add measurements values to z, and measurements positions to X.

### 6.5. Exchange Measurements (Data Fusion)

Data fusion allows robots to have a common understanding about the process of interest. In this paper we focus on multi-robot coordination strategies, and consider decentralized data fusion out of the scope of this work. Therefore, we implement a simple flooding algorithm to carry out the data fusion. Note that decentralized data fusion approaches like, e.g., [33] could also be considered. Our data fusion algorithm works works as follows: First, robots broadcast z,X to their neighbors. Second, once a robot receives z,X it will broadcast those again if this is the first time that they were received. This will continue till all robots receive measurements of the complete team.

### 6.6. Update GPs Model

Finally, robots update the GPs model with new measurements. This is done by each of the robots individually by optimizing Equation (Equation 4).

## 7. Computational Complexity

In this section, we carry out a study of the computational complexity of the proposed algorithm. We divide this study in three variants of the algorithm in order to highlight different aspects of the approach:NoCluster. This corresponds to the algorithm described in Section 6, but without considering the clustering method. That is, we consider there are as many clusters as paths resulting from the RRT for all time horizons (that is, as nodes in the RRT), where each cluster has a single path. This allows us to highlight the complexity in terms of the number of collected measurements, and total number of robots.Cluster. This is the algorithm described in Section 6. Here we highlight complexity reduction that results by introducing a clustering method.ClusterSimplified. This corresponds to algorithm described in Section 6 plus additional techniques that we introduce to reduce the computational complexity. These techniques: are kd-trees, sparse GPs [34], and the principle of *locality* [35]. Note that sparse GPs, and principle of *locality* are approximations that do not yield exact solutions. Nevertheless, these techniques have been shown to work well in practice in a large domain of problems as discussed in [34,35].

Next we analyze the worst-case computational complexity for the three aforementioned algorithm variants.

### 7.1. NoCluster

The NoCluster variant has three main components that define the algorithm’s computational complexity: (i) RRT planner, (ii) calculation of max-sum utilities, and (iii) update of the GPs model.

The complexity of RRT is given by $O(N_{p}\log N_{p})$, with $N_{p}$ the number of RRT planner iterations [26].The complexity of max-sum algorithm is determined by the calculation of $H(Y_{V_{X_{free}}}|Y_{D_{i}},Y_{N(D_{i+1:N})},X)$ in (Equation 7). This is given by the GPs regression, which is cubic on the total number of elements *m* [5] contained in $V_{X_{free}}$, $D_{i}$, $N(D_{i+1:N})$ and X. Since a robot calculates the utility of each combination of clusters (in this case, smaller or equal than $N_{p}$), the overall complexity of max-sum is $O(m^{3}N_{p}^{|N_{C}|})$, with $|N_{C}|$ the number of elements of $N_{C}≜N\left(D_{i+1:N}\right)$.The complexity of the GPs model update in (Equation 4) is given by $O(n^{3}i_{G})$, where *n* is the total number of gathered measurements, and $i_{G}$ is a user-defined parameter that sets the number of iterations we allow the optimizer to calculate the GPs hyperparameters.

The complexity of the NoCluster variant is thus determined by max-sum, as m>>n, and typically $N_{p}>>i_{G}$. The benefit of using a distributed approach such as max-sum is illustrated by noticing that the complexity scales with the number of neighbors, and not with *N*. However, it is clearly influenced by $N_{p}$, which is typically large for robots with complex dynamics, or environments with multiple obstacles. Therefore, in order to reduce the algorithm’s computational complexity we propose in this paper a concept of clustering.

### 7.2. Cluster

In the Cluster variant, the RRT structure is exploited to group $N_{p}$ nodes in $k_{s}\times k_{t}$ clusters. This yields a max-sum complexity of $O(m^{3}{\left(k_{s}k_{t}\right)}^{|N_{C}|})$. The complexity is thus now dependent on the total number of clusters, which is typically much smaller than $N_{p}$ due to the tree structure. Of course, the clustering method adds additional complexity to the algorithm. However, this is negligible [30] compared to max-sum complexity. In particular, [30] has a running time of $O(N_{p}k_{s}k_{t}d_{c}i_{c})$, with $d_{c}$ the maximum number of dimensions of a k-means state, and $i_{c}$ the number of iterations of Lloyd’s algorithm [30].

The complexity reduction of the Cluster variant is vital for an online algorithm. However, it could not be sufficient for an exploration algorithm that must run in real time. Specifically, the Cluster variant faces two main problems due to the complexity increase: (i) in max-sum algorithm as *m* grows, and (ii) in the GPs model update step as *n* grows.

### 7.3. Cluster Simplified

In order to alleviate the computational complexity of the two aforementioned problems we propose a solution that we term ClusterSimplified. On the one hand, we exploit the principle of *locality* to reduce the complexity, assuming that $x,x^{\prime }$ that are far apart are uncorrelated, and therefore do not need to be considered to carry out regression. In particular, in this work we assume that $x,x^{\prime }$ are far if $k\left(x,x^{\prime },\theta \right)<\sigma_{n}/10$. Let us point out that this is a reasonable assumption as in this paper we consider sensors with a negligible noise level. To efficiently search for locations that are correlated, we structure the data in a kd-tree.

The complexity of GPs regression is further alleviated by employing sparse GPs [34]. Specifically, we use the FITC method, with inducing points selected randomly from the set of potential measurements. Since the number of inducing points is typically set to be much smaller than the number of potential measurements, sparse GPs incur into an enormous reduction of complexity [34].

### 7.4. Summary

To finalize, we summarize in Table 1 the complexity of the three algorithm variants that we proposed in this section. Let us point out that $m_{s},n_{s}$ in Table 1 are the number of potential measurements, and actual measurements, respectively, which result after applying sparse GPs and *locality* approximations. Moreover, we analyzed in Table 2 the computation time of one algorithm run for a set of fix parameters that is representative of the simulations we carried out in the paper. In addition, we varied a set of parameters to account for several degrees of complexity reduction.

Motivated by a lower computational complexity and an equivalent performance, compared to other alternatives, we decided to employ our proposed variant ClusterSimplified in our simulations and experiments.

## 8. Simulations and Discussion of Results

### 8.1. Simulations Setup

#### 8.1.1. Generation of the Process for Exploration

We validate our algorithm in simulations in an exploration task that consists of mapping the vertical component of a wind field (see Figure 7a (By Dake (Self-made illustration) [CC BY 2.5 (http://creativecommons.org/licenses/by/2.5)], via Wikimedia Commons.)) with multiple robots. The wind field is simulated using the model proposed in [36]. The model used in [36] is an statistical model that employs data gathered from balloon and surface measurements to characterize thermals. Furthermore we added a sinusoidal component in both *x* and *y* directions to increase the complexity of the IG task. Similar modifications were done in [3]. Figure 7b depicts the resulting wind field. This corresponds to a 500 × 500 m${}^{2}$ two dimensional slice at 300 m of a three dimensional wind field. We would like to remark that the validation of our algorithm in a 2D environment (instead of in a 3D one) is motivated by two main reasons: (i) to reduce the computational complexity, and to subsequently reduce the running time of the validation in simulations, and (ii) to ease the visualization and interpretability of simulations results. The algorithm proposed in this paper is independent of the dimensionality of the environment. Therefore a 2D environment allows us to properly assess, without loss of generality, the capabilities of our algorithm to carry out an IG task.

#### 8.1.2. Robot Model

We employ a simplified aircraft model that is based on modelling discussed in [37]. We made further simplifications to adapt it to a two-dimensional environment, and assumed that the wind field does not affect the aircraft’s motion. These simplifications are still far from a realistic model. However, they allow us to demonstrate the effectiveness of the proposed IG approach.

Given these assumptions, the aircraft model is defined by the following equations:(8)$$\begin{array}{cc} x(t+\Delta t) & =x\left(t\right)+v_{in}\left(t\right)\Delta t \end{array}$$(9)$$\begin{array}{cc} \psi (t+\Delta t) & =\psi \left(t\right)+\dot{\psi }\left(t\right)\Delta t, \end{array}$$
with x(t) the aircraft’s position, $v_{in}\left(t\right)$ the aircraft’s inertial velocity, and ψ the heading angle. For airspeed *V*, commanded flight path angle θ, and commanded bank angle ϕ, the components of the velocity $v_{in}\left(t\right)=\left[v_{x},v_{y}\right]$ and $\dot{\psi }\left(t\right)$ are given by: $v_{x}=V\cos\theta \cos\psi$; $v_{y}=V\cos\theta \sin\psi$; $\dot{\psi }=\frac{g}{V}\tan\left(\phi \right)$. Let us point out that the aircraft is fully controlled by the commanded bank angle ϕ, and flight path angle θ. For the simulations we assumed an aircraft defined by the following parameters: $\Delta t=0.5\;s,\;V=15\;ms^{-1},\;g=9.8\;ms^{-2},\;\theta =0$ (constant height),ϕ∈[−π/5,π/5]rad.

#### 8.1.3. Algorithm Parameters

We consider a fleet of eight aircrafts to explore the wind field. We define a communication range $r_{c}=200$ m, a safety distance $r_{s}=10$ m, and an escape distance $r_{e}=20$ m. For the simulations we run RRT for $N_{p}=1000$ iterations, and max-sum algorithm for 5 s. For the clustering algorithm, we consider four temporal horizons at 2,5,7,10s, and three spatial divisions. This makes 12 clusters in total for each robot.

We run Monte Carlo simulations to test our approach with a number of robots that ranges between one and eight. In particular, we considered a maximum of 4 robots for the analysis in Section 8.2 and Section 8.3, and a maximum of 8 robots for the analysis in Section 8.4. The robots’ initial positions is randomly set, under the requirement that the robots network is connected. For each of the algorithms we average over 100 simulations runs. The algorithm is implemented in Python. Additionally, we use robot operating system (ROS) [38] to simulate the algorithm in a decentralized fashion.

### 8.2. Analysis of the Exploration Strategy

First we evaluate the performance of our proposed algorithm for an IG task that is not subject to constraints from Equation (Equation 1). This implies that robots run our algorithm with $U_{C}\left(\cdot \right)=0$. This algorithm variant we term it “SBMRE Alg. No Constraints”. With this study we proof the following two hypothesis:The proposed cooperation procedure, which builds on MI as information metric and max-sum as decentralized coordination technique, outperforms a benchmark algorithm.Our proposed algorithm scales as the number of robots in the system increases. That is, as the number of robots increases the performance gap between a benchmark and our algorithm grows.

To the best of our knowledge there are no algorithms in the literature that solve Equation (Equation 1); even in unconstrained form. Here we selected random walk as benchmark. A random walk have been shown to offer "surprisingly" good results for IG tasks [1,39]. Note that in this paper a random walk does not refer to the classic definition of a greedy random walk. Instead, here it refers to the generation of random trajectories. That is, a random walk implies that robots move independently following a random path, constrained by the robot motion, generated with RRT. The random walk neither aims to meet constraints nor to exchange measurements with the rest of the team. Let us remark that the random walk does not perform any data fusion, which implies that each of the robots only has measurements taken by itself. So, in order to obtain a fair comparison with our algorithm, which fuses data online, we perform a data fusion during post processing for the random walk benchmark.

Here we study our exploration strategy by evaluating the reduction of the root mean squared error (RMSE) after a 300 s exploration run. That is $RMSE\;Reduction\left[\%\right]=100\frac{RMSE(t=0)-RMSE(t=300)}{RMSE(t=0)}$. We compute the RMSE with respect to a set of $n_{G}$ points $V_{X_{free}}\in X_{free}$ that correspond to nodes of an overlaid lattice graph with a spatial resolution of 10m. We use these $n_{G}$ points to compare the difference between our estimate $\mu_{*}$, which is the result of GPs regression given z,X, and ground truth $y_{G}\left(X_{G}\right)$, with $X_{G}:=V_{X_{free}}$. This yields the following expression for the RMSE:(10)$$RMSE=\sqrt{\frac{\sum_{i=1}^{n_{G}}{\left(\mu_{*}^{\left[i\right]}-y_{G}^{\left[i\right]}\right)}^{2}}{n_{G}}}.$$

We depict in Figure 8a the RMSE reduction for one, two, three and four robots. First fact that we observe is that our algorithm offers an increase of performance with respect to a random walk of a 6% with one robot, and increases up to a 20% with four robots. Next fact is that the gap between our algorithm’s performance and a random walk increases as we add more robots to the team. According to results from Figure 8a we can confirm our two hypothesis.

### 8.3. Analysis of the Multi-Robot Coordination Strategy

We demonstrated our algorithm’s cooperation capabilities to gather information. Next we analyze our algorithm’s coordination capabilities to meet problem specific constraints from Equation (Equation 1). Therefore, here we proof two hypothesis, which correspond to the inter-robot constraints considered in this work. These are the following:Our algorithm meets the collision avoidance constraint, and outputs collision-free trajectories.The network connectivity constraint is fulfilled, and our algorithm guarantees a higher connectivity than a random walk benchmark.

#### 8.3.1. Collision Avoidance

This section evaluates the collision avoidance capabilities of our algorithm. In particular, we calculate the percentage of time that the constraint is not met during all simulation runs, which we obtained by evaluating the distance between each pair of robots for each iteration. In Figure 9 we depict one example of the inter-robot distances during one algorithm execution. Figure 9 helps us to understand how robots coordinate to avoid collisions. Moreover it illustrates a potential collision between robots around iteration number 210 and 220, as the inter-robot distance is smaller than the safety distance $r_{s}=10$ m.

For all simulation runs, the percentage of time that the collision avoidance constraint is not met is 0.16%. Let us remark here that a low percentage is still possible as the escape distance could be violated. In this sense, local safety measures and obstacle avoidance mechanisms [40] could be employed to solve such conflicts. Specifically, in [40] the authors present a collision avoidance algorithm for multiple aerial vehicle systems than functions in real-time. The proposed algorithm is based on the 3D-Optimal Reciprocal Collision Avoidance (ORCA) algorithm, and considers dynamic constraints of the UAV model and static obstacles.

A fundamental feature of our algorithm is that the violation of the collision avoidance constraint can be detected in advance by evaluating the robot individual utility function. In case of a potential collision, an algorithm like the one proposed in [40] could be executed. In contrast, a random walk has no means to anticipate a future possible collision without an external collision avoidance system.

#### 8.3.2. Network Connectivity

Next we evaluate the fulfillment of the network connectivity constraint. To this end, we calculate the percentage of iterations in which the network is not connected at the end of robots’ paths (during max-sum execution) for all simulation runs. This means that there are robots or subsets of robots that cannot communicate with the rest of the team, and therefore they violate the periodic connectivity constraint (constraint 7 in Section 3). As pointed out before, non-connectivity is an undesirable characteristic for most applications [23,41].

Figure 8b shows the network connectivity for our algorithm and a random walk. Our proposed algorithm achieves a network connectivity that ranges between 91% and 98%. In contrast, the random walk achieves a connectivity that ranges between 42% and 60%.

### 8.4. Analysis of the Algorithm’s Scalability with an Increasing Number of Robots

A fundamental aspect of any multi-robot algorithm is its scalability as the number of robots increases. In this section we analyze the scalability in terms of the computational load that each robot must deal with. As stated in Section 7, the computational load is determined by the calculation of utilities in max-sum. Specifically, computational load scales exponentially with $|N_{C}|$ – number of neighboring robots with which each individual robot must cooperate.

Therefore, we analyze $|N_{C}|$ as we increase the number of robots in the system from 2 to 8 robots. We compare our algorithm against a system that requires full connectivity of the network, or in another words, a system that is centralized. We depict in Figure 10 simulation results.

We can conclude according to Figure 10 that, in a fully connected/centralized system, $|N_{C}|$ increases linearly, which results in an exponential increase of the computational load per robot (see Table 1). In contrast, our distributed algorithm only presents a slight increase in $|N_{C}|$ as we increase the number of robots. This results in an slight increase of the computational load per robot.

To summarize: we can conclude that our algorithm scales with the number of robots as the computational load per robot only increases slightly.

### 8.5. Analysis of the Clustering Procedure

The evaluation of the exploration and coordination strategies illustrate the effectiveness of our approach to solve problem (Equation 1): performing an IG task with multiple robots while fulfilling problem specific constraints. In this section we evaluate the algorithm’s sensitivity to changes in parameters values. In particular, we focus the study on the two most relevant parameters: number of spatial-temporal clusters, and communication radius. For these two parameters we analyze: (i) the resulting RMSE between estimation and ground truth after three iterations of the algorithm, and (ii) the solution feasibility; i.e., how often the algorithm is able to find a solution that meets the constraints imposed in Equation (Equation 1).

We carry out the analysis for an environment that measures 1000×1000 square meters, with a wind field that is similar to the one shown in Figure 7b but it contains two thermals. For that scenario, we run 5000 Monte Carlo simulations with randomly chosen parameters. Specifically, the number of clusters ranges from 1 to 36, and we consider a communication radius of 200, 300, 400, 500 and 2000 m. Let us also add that we let max-sum run for 180 seconds each algorithm iteration in order to being able to calculate all utilities for up to 25 clusters.

Figure 11 depicts the results of the parameters analysis. The depicted curves are the result of a quadratic curve fitting done on the original data. From Figure 11 we can extract four main conclusions:The softer the constraints, i.e., a larger communication radius, the better the algorithm’s performance both in terms of RMSE and solution feasibility.Our algorithm’s performance increases, i.e., lower RMSE and higher solution feasibility are achieved, as we increase the number of clusters up to approximately 18 clusters. This demonstrates that the larger the number of clusters, the better we represent the original RRT, which translates into a more efficient multi-robot cooperation.The performance of the algorithm remains approximately constant for a number of clusters that ranges between 18 and 25. In another words: adding new clusters does not improve the representability of the original RRT, since clusters start containing paths that are very similar. This property leads to an enormous reduction of the algorithm’s computational complexity as indicated in Section 7.Performance of the algorithm decreases with a number of clusters greater tham 25, for our particular setup. It is essentially due to an insufficient running time for max-sum to converge, which results in a suboptimal solution. This result emphasizes the importance of point 3, since according to Figure 11 with a number of clusters equal to approximately 18, for our setup, we obtain the best performance both in terms of RMSE and solution feasibility.

This section concludes the analysis of the algorithm in simulations. Next we present experimental results.

## 9. Experiments and Discussion of Results

To validate the algorithm we carried out a field experiment with flying robots. Specifically, we explored a simulated two-dimensional wind field with quadcopters emulating a fixed-wing aircraft’s dynamics.

In this experiment we aim to proof the following statements:Our system is able to perform active IG online, according to the measured values.Our system is robust against inaccuracies in robots’ position.

Next we describe in detail the experimental setup and results.

### 9.1. Experimental Setup

#### 9.1.1. Wind Field model

We simulated a wind field, instead of measuring an actual field in order to simplify the overall experiment. This allows us to abstract ourselves from the particular sensor characteristics, and evaluate the algorithm’s performance in a real scenario.

The wind field corresponds to a scaled down version of the one described in Section 8.1 (see Figure 7). Specifically, we reduced the size of the environment by a factor 10. This results in a wind field over an area of 50×50 square meters. Here we set Δt=0.2, $r_{c}=20$ m, $r_{s}=5$ m and $r_{e}=3$ m to account for a smaller environment. For the rest of parameters, we employ the same values as in Section 8.1.

#### 9.1.2. System Architecture

To explore the afore-described wind field, we propose a system architecture that is composed of the following main elements: (i) quadcopters, (ii) a central computer, and (iii) a Real-Time Kinematic navigation for global positioning systems (GPS-RTK). Next we provide details of each of the components.

*Quadcopters.* We use three quadcopters that emulate the dynamics of a simple fixed-wing aircraft. In particular, we employ Equation (Equation 8) to plan the quadcopters trajectories. A trajectory can be represented as a set of waypoints that the quadcopters can follow using their onboard controllers. This way, quadcopters will perform a flight path that is close to the one performed by a fixed-wing aircraft. It is true that this solution does not fully emulate the dynamics of a fixed-wing aircraft. Nevertheless it is a first step towards the validation of our algorithm in a field experiment.

Figure 12a shows one of the quadcopters used for the experiment. Quadcopters are a modified version of an AscTec Hummingbird from Ascending Technologies. We equipped them with a Raspberry Pi 2 Model B that sends commands to the quadcopter’s onboard controller. Note that the core of the algorithm runs in a central computer due to the insufficient computational capabilities of a Raspberry Pi.

*Central computer.* A laptop situated outside the exploration area monitors the complete system, and runs the core of the algorithm. Specifically, it executes the algorithm that coordinates robots, and then sends waypoints to quacopters. Communication between quadcopters and the central computer is realized using Wi-Fi. Quadcopters will then fly to the commanded waypoints, using the onboard controller that runs in the Raspberry Pi. It is important to remark that the algorithm runs in a distributed fashion where each quadcopter runs in a separate software module—ROS node.

*GPS-RTK.* Quadcopters are also equipped with a GPS-RTK [42]. Specifically, we mounted the Piksi 1 modules from Swift Navigation. GPS-RTK allows us to achieve a sub-meter-level accuracy in the position.

### 9.2. Experimental Results

Here our goal is to evaluate: (i) the robots’ trajectories, (ii) the process reconstruction and the remaining uncertainty after the exploration task, and (iii) the RMSE between the process reconstruction and ground truth. For the last one we compare runs with one and three quadcopters to highlight the benefits of a multi-robot system.

#### 9.2.1. Robots Trajectories

First, we depict in Figure 12 the trajectories that quadcopters flew during the exploration run. Figure 12b corresponds to robots’ nominal position. We can observe that the shape of the trajectories resembles those of a simple fixed-wing aircraft. Moreover, robots cover the complete exploration area; except the bottom right corner. This was due to battery life constraints, which did not let robots to complete the exploration task.

Figure 12c depicts quadcopters positions as output from the GPS-RTK system. Trajectories are similar to the nominal ones. However, we observe inaccuracies in the position solution. For example, we could concentrate in Figure 12c on a large concentration of dots at coordinates x=15, y=45 meters. These dots correspond to a single commanded waypoint where a quadcopter tries to stay at. Ideally, we would like quadcopters to hold their position. However, this is not possible due to the combined effect of innacuracies in the robot’s controller, which relies on external sensors to calculate its position, and the GPS-RTK solution. Nevertheless, let us emphasize that these inaccuracies in position do not result in an inaccurate estimation of the wind field for the chosen size of the environment, as we will show next.

#### 9.2.2. Wind Field Estimation

We illustrate in Figure 13b the estimated wind field. It corresponds to the mean prediction of the GPs at each of the positions of the environment given the collected measurements. The estimated wind field can be compared to the ground truth (depicted in Figure 7b). First, we observe that estimation and ground truth are almost identical, and we can easily identify the thermal. This exemplifies the algorithm robustness to uncertainty in robot’s position. Second, we notice that the estimation is worse at those areas that were not covered during the exploration run; i.e., bottom right corner (also noticeable in Figure 13a). However, even on that area the algorithm achieves a decent reconstruction accuracy.

#### 9.2.3. Error between Estimate and Ground Truth

We show in Figure 13c an evaluation of the RMSE between estimate and ground truth resulting from the field experiment. We show curves for one and three robots running the algorithm proposed in this work. As we showed in simulations, the system with three robots achieves a much lower RMSE compared to one robot. Specifically, three robots achieve a three-fold improvement compared to one robot. This confirms the benefits, in terms of efficiency, of a multi-robot system.

## 10. Conclusions and Future Work

The paper presents an approach for multi-robot information gathering. It considers GPs as underlying model of the process to explore, information utilities for active perception, and the max-sum algorithm for multi-robot cooperation. The approach extends the state of the art by accounting for the motion constraints of the robots. This is realized through the use of motion planners such as RRT, which are able to handle such constraints. The method is able as well to handle mission team constraints such as network connectivity and collision avoidance restrictions. We achieved this by including additional terms into the utility functions in max-sum.

The whole approach is distributed, not requiring a central entity for processing. All the decision-making is decentralized, and, in our current implementation, only the data fusion component requires a broadcast mechanism at the network level (even though the system can work if the network connectivity is not fulfilled). As future work, we will consider decentralized data fusion approaches for GPs, as in [13,14], for a fully decentralized system.

The approach has been validated in simulation for the exploration of a wind field. We have also tested the methods in experiments with robots for the same application. The results show how the cooperation allows for a more efficient exploration, more evident when the number of robots grow. Furthermore, the results show how the approach can handle constraints that are relevant for real scenarios, in particular maintaining the network connectivity in the fleet.

One of the limitations of the presented application is the use of a fixed chain network topology, which constraints the ability of the fleet to explore. More dynamic and flexible network topologies would definitely allow for better information gathering efficiency. Furthermore, the connectivity model is a simplification, and more details regarding the actual physical layer and communication technology would be needed to model the communication constraints. Please notice that this is not a restriction of the decision making method itself.

The vehicle models employed in this work are a simplified version of a fixed-wing aircraft. We plan to extend those models to full 3D models that consider also aerodynamic effects, as a next step to apply the approach for the autonomous soaring of gliders. We will consider exploration in 3D, and combining the exploration techniques presented with the exploitation of the wind information for longer endurance of the flight. Exploitation terms can be easily included into our utility functions. The analysis of different combinations and weighting of the terms that compose the utility function is also a venue for future work. One additional aspect to consider will be the effect of the uncertainties in the wind field over the finally executed paths, and how to include this also into the constraints of the system.

## Figures and Tables

**Figure 1 sensors-20-00484-f001:**
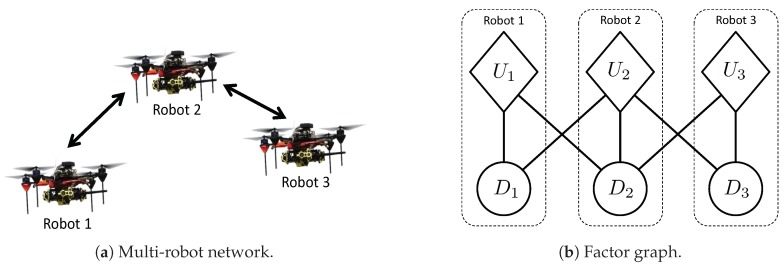
Left: a multi-robot network, where arrows depict inter-robot communication. Right: a factor graph representing utility function $U\left(D\right)=U_{1}\left(D_{1},D_{2}\right)+U_{2}\left(D_{1},D_{2},D_{3}\right)+U_{3}\left(D_{2},D_{3}\right)$. Diamonds correspond to factors, and circles correspond to variables. Note that dependencies between factors and variables are determined by the multi-robot communication network (Figure 1a).

**Figure 2 sensors-20-00484-f002:**
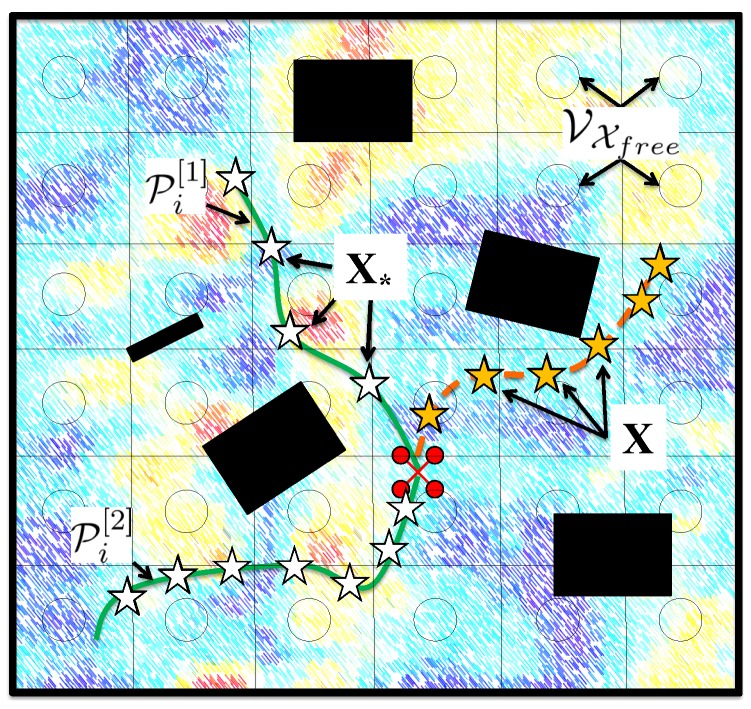
Graphical representation of the notation employed in this paper. We depict an scenario in which a robot *i* (colored red) aims to explore a process (in the background) in an environment populated with obstacles (colored black). Orange stars correspond to measurements that were previously gathered by the robot at positions X. White stars are potential measurements locations $X_{*}$. As in this chapter we consider a path planning mechanism, $X_{*}$ belong to potential paths $P_{i}^{\left[1\right]}$, $P_{i}^{\left[2\right]}$ that could be traversed by the robot. Information metrics are utilized here to quantify the informativeness of potential paths. In addition, we also represent $V_{X_{free}}$, together with associated grid cells, which are needed to compute some metrics.

**Figure 3 sensors-20-00484-f003:**
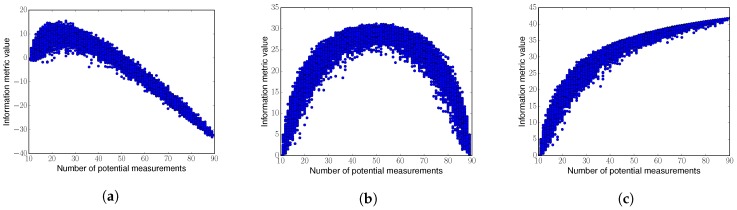
Evaluation of several information metrics as we increase the number of potential measurements. (**a**) Differential Entropy; (**b**) Mutual Information Non-Measured; (**c**) Mutual Information All. Figure 3c corresponds to our proposed metric.

**Figure 4 sensors-20-00484-f004:**
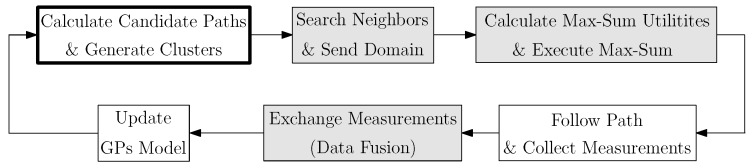
Algorithm block diagram. Shadowed blocks represent modules that require communication between robots.

**Figure 5 sensors-20-00484-f005:**
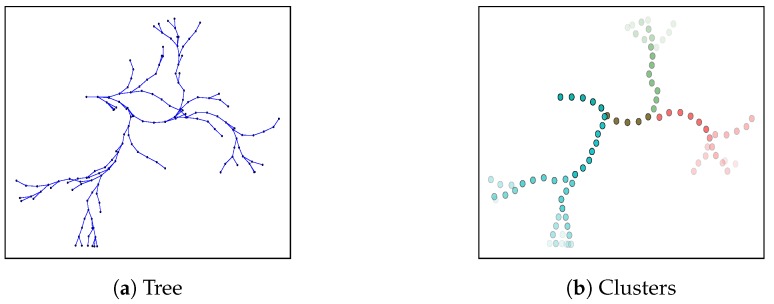
Spatio-temporal clustering. On the left hand side we depict a Rapidly exploring Random Trees algorithm (RRT). On the right hand side we depict the clusters calculated with our proposed clustering procedure for the RRT. Specifically, we considered one temporal horizon and three spatial clusters; i.e., $k_{t}=1,\;k_{s}=3$, respectively. Each of the colors represent a spatio-temporal cluster.

**Figure 6 sensors-20-00484-f006:**
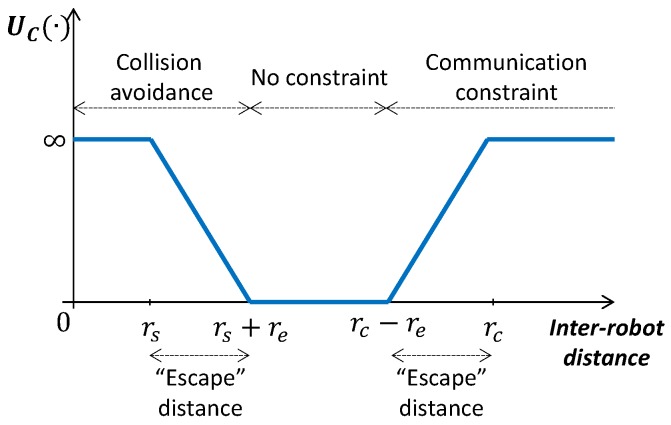
Graphical illustration of $U_{C}\left(\cdot \right)$.

**Figure 7 sensors-20-00484-f007:**
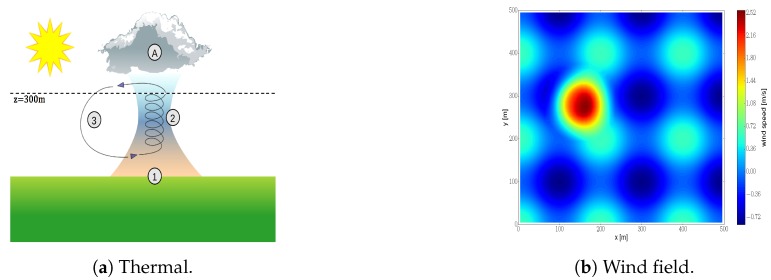
Illustration of a thermal (**a**) and the two dimensional wind field to be explored (**b**).

**Figure 8 sensors-20-00484-f008:**
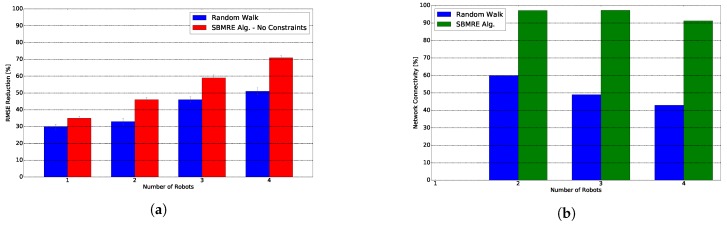
Root Mean Squared Error (RMSE) reduction and network connectivity. (**a**) RMSE reduction during an exploration task as we increase the number of robots in the system. (**b**) Percentage of iterations in which the network fulfills a periodic connectivity constraint.

**Figure 9 sensors-20-00484-f009:**
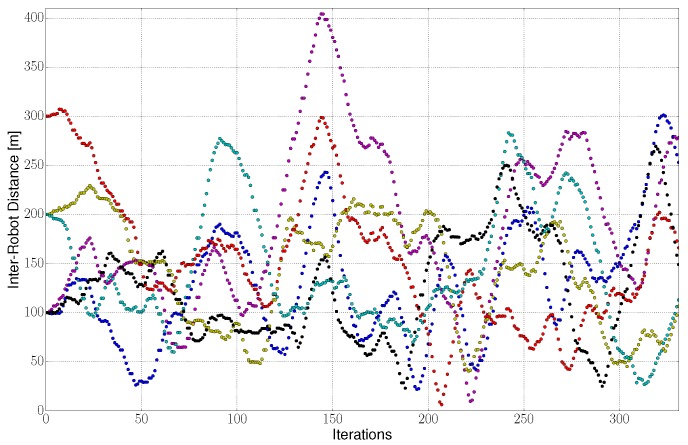
Inter-robot distance during one illustrative execution of our algorithm. Each color represents the distance between a different pair (6 different pairs) of robots for a 4-robot team.

**Figure 10 sensors-20-00484-f010:**
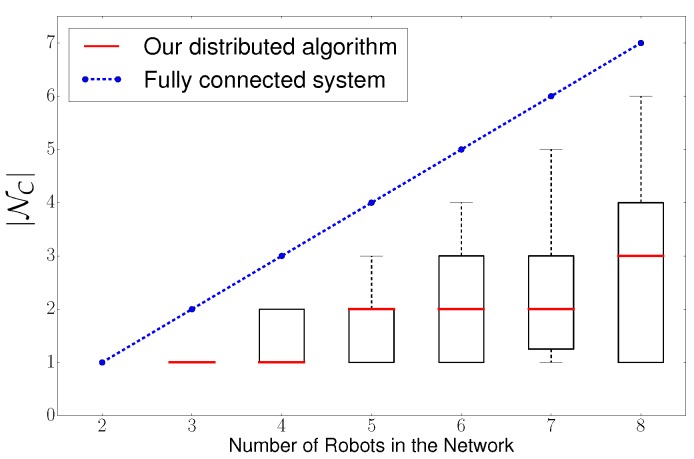
Algorithm’s scalability for an increasing number of robots. Scalability is measured in terms of the computational load per robot, which is exponential in $|N_{C}|$. We compare a fully connected/centralized system against our distributed solution.

**Figure 11 sensors-20-00484-f011:**
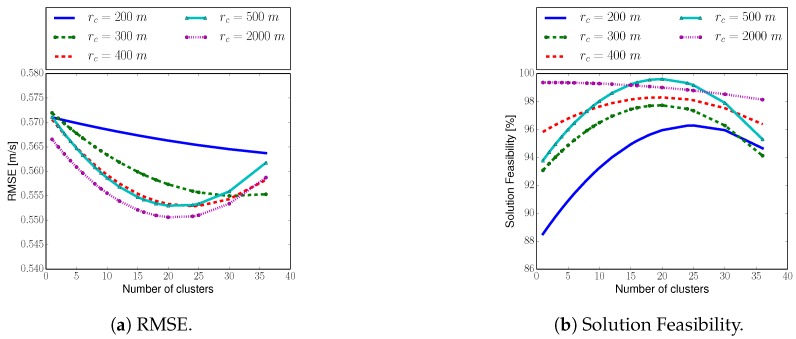
Algorithm’s performance as we vary the two most relevant parameters: number of spatial-temporal clusters, and communication radius. For these two parameters we analyze: (**a**) the resulting RMSE, calculated with (Equation 10), after three iterations of the algorithm, and (**b**) the solution feasibility; i.e., how often the algorithm finds a solution that meets constraints from Equation (Equation 1).

**Figure 12 sensors-20-00484-f012:**
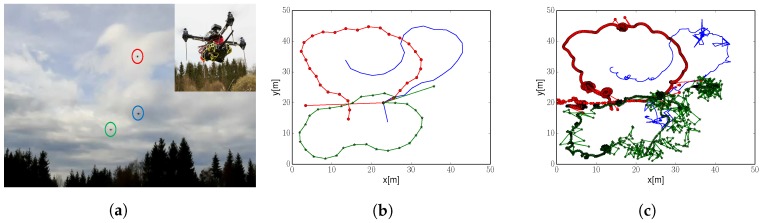
Nominal position and actual position of the quadcopters during the exploration task. Each of the three quadcopters is represented with a different color. (**a**) Quadcopters during the experiment. (**b**) Nominal position. (**c**) Actual position.

**Figure 13 sensors-20-00484-f013:**
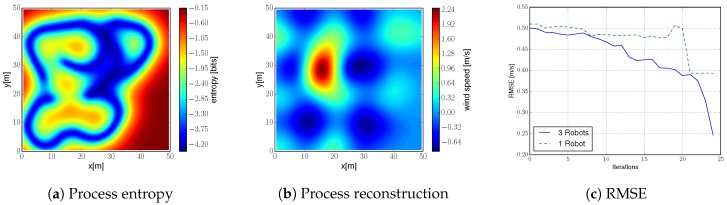
(**a**,**b**) Process entropy and reconstruction after performing an exploration run. (**c**) RMSE between estimate and ground truth for one and three quadcopters.

**Table 1 sensors-20-00484-t001:** Evaluation of algorithm’s complexity. For clarification, let us add that typically $m_{s}<<m$, $k_{s}k_{t}<<N_{p}$, and $n_{s}<<n$.

	NoCluster	Cluster	ClusterSimplified
Path planner	$O(N_{p}\log N_{p})$	$O(N_{p}\log N_{p})$	$O(N_{p}\log N_{p})$
Clustering method	-	$O(N_{p}k_{s}k_{t}d_{c}i_{c})$	$O(N_{p}k_{s}k_{t}d_{c}i_{c})$
Max-sum	$O(m^{3}N_{p}^{|N_{C}|})$	$O(m^{3}{\left(k_{s}k_{t}\right)}^{|N_{C}|})$	$O(m_{s}^{3}{\left(k_{s}k_{t}\right)}^{|N_{C}|})$
Updating GPs	$O(n^{3}i_{G})$	$O(n^{3}i_{G})$	$O(n_{s}^{3}i_{G})$

**Table 2 sensors-20-00484-t002:** Computational time required to execute one algorithm run. For the calculations, we used a set of fix parameters that is representative of the simulations we carried out in the paper. These parameters are as follows: $N_{p}=1000$, $d_{c}=20$, $i_{c}=10$, $i_{G}=10$, m=2500, $|N_{C}|=7$. In addition, we varied parameters $m_{s}$, $k_{s}k_{t}$, $n_{s}$ to account for several degrees of complexity reduction. In particular, we considered: $\left[m_{s},k_{s}k_{t},n_{s}\right]=\left[m,N_{p},n\right]/10,\;\left[m,N_{p},n\right]/50,\;\left[m,N_{p},n\right]/100,\;\left[m,N_{p},n\right]/200$. The set of parameters used for our simulations corresponds to the ClusterSimplified with a reduction factor of 100. Execution time for this set of parameters is approximately 6 s.

	NoCluster	Cluster			ClusterSimplified			
		/10	/50	/100	/10	/50	/100	/200
Computational time [s]	3 × $10^{20}$	3 × $10^{13}$	4× $10^{8}$	3× $10^{6}$	3× $10^{10}$	3075	6	3

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
