# Peer review of "Distributed Multi-Robot Information Gathering under Spatio-Temporal Inter-Robot Constraintsâ€"

_sensors, 2020, doi:10.3390/s20020484_

Round 1
Reviewer 1 Report
This is a very complete work in which it is addressed in a theoretical way and with subsequent experimental validation, the development of algorithms for the capture of data by UAV fleets in a 3D environments where the variable to be measured evolves spatially.
The algorithm is composed of a series of modules that include the proposal of candidate trajectories for each UAV through a variation of the RRT and its clustering to reduce the computational cost, the exchange of the possible trajectories with the nearby UAVs to assign cooperatively the space to be covered by each UAV, the capture of data itself and the sharing of these to reconstruct the spatial map of the measured variable.
The article has a lot of information being sometimes difficult to read, but it is well structured, with an adequate definition of the problem and a correct review of previous works, it also includes a background section, highly recommended given the amount of algorithmic techniques used in the job.
After the sequential description of the algorithms that compose the proposed method, including an evaluation of the computational cost, a validation in simulation and an experimental test in a controlled environment in which the spatial variation of the variable to be measured is simulated is approached. In these, the scenario is reduced to 2D, avoiding the quantification of the loss of generality and the implications of the extension to 3D.
As the authors clarify at the beginning, the paper is an extended version of another published in the ICRA 2018. Comparing both papers it is observed that, although the same algorithms are described in both, really, in the one presented to Sensors, the description is considerably more detailed and in that sense this new work has significant contributions for the reader.
As a whole, the article is suitable for publication and only minor comments are made, although it is noted that, due to the high number of sub-algorithms that it uses and its complexity, it is, as a whole, difficult to read.
Minor comments:
In Equation (5) the utility function is defined as the difference between the Information Gathering Utility Ui (D, X) and the Constrain Satisfaction Utility Uc (D), giving both the same weight. Couldn't it be interesting to explore the possibility of weighing both functions with different coefficients according to the scenario and the capabilities of UAVs? In any case this does not modify the development at all, since only the way in which the utility function U is calculated is altered.
In section 8 it is not clear if the simulations are in a 2D or 3D scenario. From the condition that the flight path angle (Theta) is 0 it follows that it is 2D, but it should be clarified in the scenario definition. In the same way, a reflection should be made on the implications that this simplification entails in the validation of the aspects dealt with at the theoretical level in the previous sections.
Also in section 8.1.3 it is indicated that 8 aircrafts are considered, but some results (figure 8 and line 536) are limited to 4. Can you clarify this please?
How do you decide the right number of robots for the mission? The chosen number (8 , or 4?) seems arbitrary and can certainly affect the result. For example: given the workspace in the simulation (500x500 m2) the collision probabilities are low for 8 robots. It would be convenient to justify the number of UAVs chosen and evaluate the implications of using a larger number, even with some simulation.
In the actual experiment (section 9) the workspace is scaled to 50x50m2. To maintain proportionality with the simulations (section 8), shouldn't the collision, escape and range of communications distances also been scaled?
A minor comment is also made on the contributions of the authors, referenced at the end of the paper. It indicates the contribution of A. Viseras and L. Merino, but no contribution from the second author Z.Xu.
Author Response
This is a very complete work in which it is addressed in a theoretical way and with subsequent experimental validation, the development of algorithms for the capture of data by UAV fleets in a 3D environments where the variable to be measured evolves spatially.
The algorithm is composed of a series of modules that include the proposal of candidate trajectories for each UAV through a variation of the RRT and its clustering to reduce the computational cost, the exchange of the possible trajectories with the nearby UAVs to assign cooperatively the space to be covered by each UAV, the capture of data itself and the sharing of these to reconstruct the spatial map of the measured variable.
The article has a lot of information being sometimes difficult to read, but it is well structured, with an adequate definition of the problem and a correct review of previous works, it also includes a background section, highly recommended given the amount of algorithmic techniques used in the job.
After the sequential description of the algorithms that compose the proposed method, including an evaluation of the computational cost, a validation in simulation and an experimental test in a controlled environment in which the spatial variation of the variable to be measured is simulated is approached. In these, the scenario is reduced to 2D, avoiding the quantification of the loss of generality and the implications of the extension to 3D.
As the authors clarify at the beginning, the paper is an extended version of another published in the ICRA 2018. Comparing both papers it is observed that, although the same algorithms are described in both, really, in the one presented to Sensors, the description is considerably more detailed and in that sense this new work has significant contributions for the reader.
As a whole, the article is suitable for publication and only minor comments are made, although it is noted that, due to the high number of sub-algorithms that it uses and its complexity, it is, as a whole, difficult to read.
First of all, we would like to thank the reviewer for the effort devoted to review the paper, and for the positive and encouraging comments.
We agree with the reviewer that the paper might be sometimes difficult to read, as it requires a deep understanding of multiple complex techniques. Nevertheless, we think that the proposed structure helps readers to sequentially and slowly dive into the paper. In this respect, we highly appreciate the positive comments from reviewer regarding the paper’s structure.
Next we address the subsequent questions raised by the reviewer. We really hope that our response, together with the modifications we carried out in the paper, will contribute to ease the reading and understanding of the manuscript.
Minor comments:
In Equation (5) the utility function is defined as the difference between the Information Gathering Utility Ui (D, X) and the Constrain Satisfaction Utility Uc (D), giving both the same weight. Couldn't it be interesting to explore the possibility of weighing both functions with different coefficients according to the scenario and the capabilities of UAVs? In any case this does not modify the development at all, since only the way in which the utility function U is calculated is altered.
Yes, thanks a lot for the suggestion. Definitely it is interesting to analyze different weightings, which would affect how close we allow the robots to “approach” the constraints, and this could depend on the capabilities on the robots.
As mentioned, the max-sum algorithm would work just the same, even though it may affect the iterations needed to converge. We leave the analysis of the structure of the utility function for future work. In Section 10, we included some text to reflect that this point is one of the promising venues for future work.
In section 8 it is not clear if the simulations are in a 2D or 3D scenario. From the condition that the flight path angle (Theta) is 0 it follows that it is 2D, but it should be clarified in the scenario definition. In the same way, a reflection should be made on the implications that this simplification entails in the validation of the aspects dealt with at the theoretical level in the previous sections.
Thanks for this observation. Indeed, the scenario used for simulations is a 2D scenario, as pointed out by the reviewer. We fully agree with the reviewer that this point was not correctly addressed in the paper. In particular, the implications of simulating a 2D scenario, instead of a 3D one, were not addressed.
First of all, we would like to stress that the method proposed in the paper is independent of the dimensionality of the scenario and of the robot’s state space. We decided to validate our algorithm in a 2D scenario for two reasons: first, for visualization and interpretability purposes, as a 2D scenario is much easier to visualize and to interpret; and second, for computational reasons, as the computational complexity of the algorithm would increase with the dimensionality of the scenario following the analysis presented in Section 7. The latter implies a much longer computational time to carry out the multiple Montecarlo simulations required to validate the algorithm.
Since the validation of the algorithm in a 2D scenario does not violate any of the assumptions of the algorithm, we decided, without loss of generality, to consider this option. We modified the text in section 8.1.1 to comment on the implications of choosing a 2D scenario for validation.
Also in section 8.1.3 it is indicated that 8 aircrafts are considered, but some results (figure 8 and line 536) are limited to 4. Can you clarify this please?
Thanks for pointing out this issue, as the manuscript in its current form is confusing for the reader. We fully agree with the reviewer.
For the analysis in Sections 8.2 and 8.3, which focus on the analysis of the exploration and coordination strategies, respectively, we decided to simulate up to 4 robots as this was sufficient to illustrate the advantages of our algorithm with respect to the considered benchmarks. Then, in Section 8.4, our objective was to specifically target the analysis of our algorithm’s scalability as the number of robots in the system increases. To this end, we increased the number of robots from 1 to 8 to better illustrate the scalability.
To avoid misunderstandings related to the number of robots used in the validation, we modified the text in Section 8.1.3 to clearly specify the number of robots used in each of the simulations we carried out in the paper.
How do you decide the right number of robots for the mission? The chosen number (8 , or 4?) seems arbitrary and can certainly affect the result. For example: given the workspace in the simulation (500x500 m2) the collision probabilities are low for 8 robots. It would be convenient to justify the number of UAVs chosen and evaluate the implications of using a larger number, even with some simulation.
As mentioned above, the use of 4 and up to 8 vehicles in the simulations was selected for purposes of evaluating different aspects of the approach, like scalability, etc. and not actually as a function of the mission.
We agree with the reviewer that it is interesting to have some guidelines to determine the right number of robots for a given mission, and to consider the implications of using larger number of robots. In this regard, in the paper we analyze the impact on the RMSE of the reconstructed process, showing how the percentage of RMSE reduction increases with the number of robots. This would be a good indicator to decide the required number of robots. At the same time, the computational power required per robot increases with the size of the team (as per Fig. 9), and the connectivity constraint fulfillment slightly degrades.
In this respect, we strongly believe that the results presented in the paper justify the use of the number of robots needed to carry out the specific IG task that we considered in the paper. Of course, providing a general answer to arbitrary IG tasks is very interesting, but it is a research question by itself and we consider it as part of the future work.
In the actual experiment (section 9) the workspace is scaled to 50x50m2. To maintain proportionality with the simulations (section 8), shouldn't the collision, escape and range of communications distances also been scaled?
The reviewer is correct. We completely forgot to indicate how we scaled the mentioned parameters. Thanks for pointing it out.
In particular, we scaled the communication distance (r_c) by a factor 10 and set it to 20 meters, the scape distance (r_e) to 3 meters, and the collision distance (r_s) to 5 meters to guarantee no collisions between UAVs. We now modified the text in Section 9.1.1 to clarify this aspect.
A minor comment is also made on the contributions of the authors, referenced at the end of the paper. It indicates the contribution of A. Viseras and L. Merino, but no contribution from the second author Z.Xu.
Thank you very much for this comment. We somehow forgot to update the author contributions. The reviewer can find the updated author contributions in the revised version of the manuscript.
Reviewer 2 Report
I can see the potential of your overall work. I a good approach towards a way to add the complex robot dynamics into previously defined idealized models. Sections 1-6 nicely described the purpose of your work. However, in section 7 I would have like to see some work comparing the cost of each algorithm in Table 1. Some tabulation or analysis of the computing power it takes for the robot to find a solution. In this cases, Game Theory is a good place to explore similar work on path planning or clustering. I have seen a significant reduction of computing power and an enhance capability for robots to change paths on the fly due to external constraints such as water current on surface unmanned vehicles. But I do give you credit into doing all the needed literature review and identify how to enhance existing models in order to deal with uncertainty. In section 8.3.1, I feel that it lacks rigor. Collision avoidance is highly important in multi-agents and describing to percentage values doesn't give enough robustness to your work. However, adding a Table, of graphs showing more significant data can help. Or you can explain how further work will be done not just by adding a reference but adding more explanation. For 8.3.2, to truly analyze connectivity there needs to be a data packet analysis. Need more detail on what king of communication: Wi-Fi, ZigBee, LoRa, etc. How your drones will communicate and how would you verify the data packet transmission. The clustering part is well explained, and it shows promise. For the actual experiments you didn't explain if the Astec drone can truly emulate a fix-wing. For example, the DJI Mavic Pro has a fix-wing mode that it truly emulates the dynamics and kinematics of an RC plane. But how did you accomplish that on your Astec system. that needed more explanation. However, Figure 12 showed good promise a meaningful improvement by having more drones interact with each other.
I have the feeling that you tried to accomplish many tasks when you needed to put more rigor on one at a time. Possibly the clustering part and the GP parts can be the strongest elements on your work. I am looking forward to the further development of your ideas.
Note: there is some issues in the consistency of your text in small thing like describing "Figure vs. Fig". That is more esthetic and easier to fix. Also, I don’t know if you are using Latex for your text, but it is easier to follow your text if the Figures are located after they have been introduce in the text. I always the headache of having to chase around figures back and for. See if you can improve the flow of your text and their figures.
Author Response
I can see the potential of your overall work. I a good approach towards a way to add the complex robot dynamics into previously defined idealized models. Sections 1-6 nicely described the purpose of your work. However, in section 7 I would have like to see some work comparing the cost of each algorithm in Table 1. Some tabulation or analysis of the computing power it takes for the robot to find a solution.
Thanks for the positive comments. Regarding the comment about section 7, we agree with the reviewer that a more tangible summary of the computational cost was missing in the paper. To this end, we added Table 2 in Section 7 to analyze the computation time of one algorithm run for a set of fixed parameters that is representative of the simulations we carried out in the paper. In addition, we varied a set of parameters to account for several degrees of complexity reduction.
The summary provided in Table 2 exemplifies the importance of the ClusterSimplified method. We now hope that complexity results are easier to interpret.
In this cases, Game Theory is a good place to explore similar work on path planning or clustering. I have seen a significant reduction of computing power and an enhance capability for robots to change paths on the fly due to external constraints such as water current on surface unmanned vehicles. But I do give you credit into doing all the needed literature review and identify how to enhance existing models in order to deal with uncertainty.
We would like to thank the reviewer. We focused our analysis of the state of the art on the exploration of GP models by multiple robots. In particular, we model the problem as a Distributed Constraint Optimization problem. Game-theoretic methods are also a model for addressing the problem of multi-robot decision-making, including cases with imperfect information. We have not found any approach to this particular problem (exploration of GP models) employing game-theoretic reasoning.
We modified the text in Section 1, and extended a bit the state of the art to include references in this regard. Here we would like to point out a relevant reference, which we now included in the paper, and links game theory and multi-robot decision making:
“Oliehoek, F. A., Whiteson, S., & Spaan, M. T. (2012, August). Exploiting structure in cooperative Bayesian games. In Proceedings of the Twenty-Eighth Conference on Uncertainty in Artificial Intelligence (pp. 654-663). AUAI Press.”
In addition, Bayesian-Games are also used for a multi-robot decision making task and DCOP (in particular, Max-Sum) is used to solve it under certain conditions.
In section 8.3.1, I feel that it lacks rigor. Collision avoidance is highly important in multi-agents and describing to percentage values doesn't give enough robustness to your work. However, adding a Table, of graphs showing more significant data can help. Or you can explain how further work will be done not just by adding a reference but adding more explanation.
Thanks for this comment. We agree with the reviewer that inter-robot collision avoidance is a fundamental aspect in multi-robot systems, and it was not properly addressed in the paper.
Following reviewer’s suggestions, we introduced two modifications in the manuscript. First, we run our algorithm and plotted an example of the inter-robot distances during one illustrative execution of the algorithm. This is depicted in Figure 9 and described in Section 8.3.1 in the revised version of the manuscript. Figure 9 helps us to understand how robots coordinate to avoid collisions. In addition, it also illustrates a potential collision between robots around iteration number 210 and 220 (inter-robot distance smaller than the safety distance r_s=10 meters).
Second, we included in Section 8.3.1 a further explanation about how local safety measures and obstacle avoidance mechanisms could be employed to solve potential collision conflicts.
We really hope that the included material properly addresses reviewer’s concerns.
For 8.3.2, to truly analyze connectivity there needs to be a data packet analysis. Need more detail on what king of communication: Wi-Fi, ZigBee, LoRa, etc. How your drones will communicate and how would you verify the data packet transmission.
Here in the paper, as in other approaches in the literature, we restrict our analysis to considering constraints coming from a simplified radius coverage model regarding the minimum distance required to create a link.
We agree in that the connectivity requires taking into account other characteristics of the physical layer and reasoning about additional levels in the communication stack. However, the method presented allows us to take into account additional constraints coming from those other considerations, like the need to maintain communication for a given time. We consider this aspect as future work, and we have modified the discussion section in this regard.
For the experiments, quadcopters used Wi-Fi for communication. This detail we forgot to mention it in the paper. We now modified the text in Section 9.1.2 to specify this aspect.
The clustering part is well explained, and it shows promise.
Thank you very much for the positive comment.
For the actual experiments you didn't explain if the Astec drone can truly emulate a fix-wing. For example, the DJI Mavic Pro has a fix-wing mode that it truly emulates the dynamics and kinematics of an RC plane. But how did you accomplish that on your Astec system. that needed more explanation.
We would like to thank the reviewer for the comment. In fact, the Asctec drone cannot truly emulate a fix-wing, in the sense a DJI Mavic Pro does. To overcome this issue we did the following: we planned trajectories considering the dynamics of a fixed-wing. Then we defined a trajectory as a list of waypoints, which we send to the Asctec onboard controller. It might not be the “perfect” solution, but it allows us to validate our proposed algorithm in a field experiment. In this direction, our next steps include a more realistic setting like the one suggested by the reviewer using a DJI Mavic Pro, followed by experiments with actual fix-wing aircraft.
To clarify the issue raised by the reviewer, we added a note in Section 9.1.2.
However, Figure 12 showed good promise a meaningful improvement by having more drones interact with each other.
Thanks for the positive comment.
I have the feeling that you tried to accomplish many tasks when you needed to put more rigor on one at a time. Possibly the clustering part and the GP parts can be the strongest elements on your work. I am looking forward to the further development of your ideas.
Thanks for this comment. In this point, we must partially disagree with the reviewer. In the paper we proposed a solution to accomplish “distributed multi-robot information gathering under spatio-temporal inter-robot constraints”. In order to come up with a solution to this problem, we had to introduce multiple complex techniques. This has as a consequence that an extensive theoretical background is required, which might make the paper hard to understand and lose the reader’s focus.
After introducing reviewers’ suggestions, we now think that the paper is more rigorous, and easier to understand for the reader. Additionally, we also hope that the paper’s goal and justification of the proposed methods became clear in the revised version of the manuscript.
Note: there is some issues in the consistency of your text in small thing like describing "Figure vs. Fig". That is more esthetic and easier to fix.
Thanks for the suggestion. This is definitely crucial to have a consistent text. We went through the paper and modified it accordingly.
Also, I don’t know if you are using Latex for your text, but it is easier to follow your text if the Figures are located after they have been introduce in the text. I always the headache of having to chase around figures back and for. See if you can improve the flow of your text and their figures.
Thanks for this comment. We agree with the reviewer that a correct placement of figures really helps to follow the paper flow. In this respect, we followed reviewer’s suggestion and tried to position figures on top of the page in which they are referenced. It is however not always possible due to the large amount of figures we have in the paper.
Nevertheless, we hope that the paper is now easier to follow.
Round 2
Reviewer 2 Report
You did a good job working on the recommendations to improve your paper. I think is ready for publication now. I am looking forward to the evolution of your research.
Author Response
Thank you very much for the time devoted to review our work. We highly appreciate it.
We are very pleased that our answers satisfactorily addressed all your concerns.